# Addictions, Social Deprivation and Cessation Failure in Head and Neck Squamous Cell Carcinoma Survivors

**DOI:** 10.3390/cancers15041231

**Published:** 2023-02-15

**Authors:** Lara Nokovitch, Yonjae Kim, Philippe Zrounba, Pierre-Eric Roux, Marc Poupart, Rabiha Giagnorio, Dominique Triviaux, Charles Maquet, Jennifer Thollin, Nathalie Arantes, Nathalie Thomas, Béatrice Fervers, Sophie Deneuve

**Affiliations:** 1Surgical Oncology Department, Centre Léon Bérard, UNICANCER, 69008 Lyon, France; 2Medical Oncology Department, Centre Léon Bérard, UNICANCER, 69008 Lyon, France; 3ENT Department, CHU de Rouen, 76000 Rouen, France; 4Prevention Cancer Environment Department, Centre Léon Bérard, 28 Rue Laënnec, 69008 Lyon, France; 5Inserm, U1296 Unit, “Radiation: Defense, Health and Environment”, Centre Léon Bérard, 28 Rue Laënnec, 69008 Lyon, France

**Keywords:** head and neck neoplasms, tobacco use cessation, alcohol abstinence, social deprivation, cancer survivors

## Abstract

**Simple Summary:**

Tobacco and alcohol are well-known risk factors of head and neck squamous cell carcinoma (HNSCC). A low socio-economic status also represents an independent risk factor of HNSCC. However, tobacco, alcohol, and social precariousness are rarely assessed by oncologists. The aim of this prospective study was to evaluate the evolution of tobacco/alcohol consumption and dependence, as well as social deprivation, in survivors of a first HNSCC to whom systematic screening and management were proposed from the time of diagnosis. We show that addictions and social deprivation tend to improve when taken care of from the diagnosis. The most dependent and frail patients are at highest risk of cessation failure. Intervention targeting smoking and drinking at the same time might be more effective. Smoking and drinking cessation might improve survival for HNSCC patients.

**Abstract:**

Aim: To evaluate the evolution of addictions (tobacco and alcohol) and social precarity in head and neck squamous cell carcinoma survivors when these factors are addressed from the time of diagnosis. Methods: Addictions and social precarity in patients with a new diagnosis of HNSCC were assessed through the EPICES score, the Fagerström score, and the CAGE questionnaire. When identified as precarious/dependent, patients were referred to relevant addiction/social services. Results: One hundred and eighty-two patients were included. At the time of diagnosis, an active tobacco consumption was associated with alcohol drinking (Fisher’s exact test, *p* < 0.001). Active smokers were more socially deprived (mean EPICES score = mES = 36.2 [±22.1]) than former smokers (mES = 22.8 [±17.8]) and never smokers (mES = 18.9 [±14.5]; Kruskal-Wallis, *p* < 0.001). The EPICES score was correlated to the Fagerström score (Kruskal-Wallis, *p* < 0.001). Active drinkers (mES = 34.1 [±21.9]) and former drinkers (mES = 32.7 [±21]) were more likely to be socially deprived than those who never drank (mES = 20.8 [±17.1]; Krukal-Wallis, *p* < 0.001). A Fagerström score improvement at one year was associated to a CAGE score improvement (Fisher’s exact test, *p* < 0.001). Tobacco and alcohol consumption were more than halved one year after treatment. Patients who continued to smoke one year after diagnosis were significantly more likely to continue to drink (Fisher’s exact test, *p* < 0.001) and had a significantly higher initial EPICES score (Kruskal-Wallis, *p* < 0.001). Conclusions: At one year, addictions and social deprivation tend to improve when taken care of from the diagnosis. The most dependent patients and those with multiple frailties are at highest risk of cessation failure.

## 1. Introduction

Tobacco and alcohol are well-known risk factors of head and neck squamous cell carcinoma (HNSCC) [1]. Among HNSCC patients, cigarette smoking is often associated with alcohol consumption [2]. About 25% of HNSCC survivors will develop a second primary cancer [3], continuing alcohol and tobacco intoxication being major risk factors for recurrence and second primary occurrence [3,4]. Smoking is also associated to poorer survival [5], whereas the association between alcohol drinking and survival of individuals with HNSCC remains unclear [6,7]. Thus, cessation of tobacco smoking and alcohol drinking is a major concern in the treatment of HNSCC. However, the success of quitting attempts among tobacco smokers is correlated to the degree of nicotine dependence [8].

Apart from these two major risk factors, a low socio-economic status (SES) represents an independent risk factor, comparable to tobacco and alcohol consumption, that is associated with an increased risk of HNSCC across the world [9,10]. A low SES is also frequently associated with smoking and alcohol consumption [10], and increases the risk of multimorbidity, frailty, and disability in HNSCC patients [11]. However, social deprivation is insufficiently taken into account in studies exploring smoking and alcohol cessation in HNSCC patients [12,13].

Tobacco and alcohol are modifiable risk behaviors that can be targeted from the time of HNSCC diagnosis [14], with the possibility to offer patients an opportunity to enter a tobacco/alcohol cessation program [14]. The timing of when clinical tobacco intervention is offered has been identified as a critical factor in smoking cessation efforts, with lower smoking relapse rates in HNSCC patients who quit prior to receiving surgery [13]. However, a recent survey suggests that fewer than half of oncologists actually offer assistance with smoking cessation [15], cancer care providers’ representation of addictions being mostly focused on the pathology and cancer treatment [16]. Thus, the diagnosis of HNSCC remains under-used as a teachable moment for alcohol and smoking cessation [14].

According to the Head and Neck 5000 study [12], people who both drink and smoke are less likely to change these behaviors and are likely to benefit the most from effective interventions targeting both behaviors. A systematic review [17] evaluating three randomized controlled trials of smoking and alcohol cessation interventions in patients with HNSCC and oral dysplasia also showed that interventions targeting both behaviors tended to be more effective in reducing smoking prevalence than interventions targeting smoking only. To date, only Duffy et al., in a randomized study [18], explored smoking and alcohol cessation in HNSCC patients after the implementation of a tailored intervention targeting smoking, alcohol, and depression.

The aim of this prospective study was to evaluate the evolution of tobacco/alcohol consumption and dependence, as well as social deprivation, in survivors of a first HNSCC to whom systematic screening and management of these three conditions were proposed from the time of diagnosis.

## 2. Materials and Methods

A prospective monocentric observational cohort study was conducted between January 2017 and December 2019. The study complied with the General Data Protection Regulation (GDPR registration number: R201-004-108) of the Regional Comprehensive Cancer Center Léon Bérard (CLB), Lyon, France, and was approved by the local Ethics Committee.

All patients with newly diagnosed first HNSCC for whom surgical treatment was planned were eligible to enter the study. During the first consultation before surgery, the surgeon proposed and explained the study to the patient.

Sociodemographic and clinical data including gender, date of birth, anthropometrics (body mass index (BMI), calculated as the body weight (kilogram) divided by the square of the height (centimeters)), and socio-economic and employment status were collected face-to-face at baseline by a nurse coordinator after the multidisciplinary tumor board decision. All clinical data were extracted from the patients’ electronic medical records.

Data on patients’ addictions and social deprivation (tobacco and alcohol consumption and dependence and social deprivation) were collected face-to-face by the nurse at baseline (T0), at 6 months (T1), and at 12 months (T2) during follow-up consultations. The number of pack-years was evaluated for current and former smokers. Current smokers were defined as smoking at least one cigarette per day. Former smokers were defined as having quit smoking for more than one month at baseline. Patients were considered never-smokers if they had smoked <100 cigarettes in their lifetime [19]. Alcohol average daily intake was evaluated for current and former drinkers. Current drinkers were patients with an alcohol consumption exceeding the World Health Organization (WHO) recommended number of drinks per week [20]. Former drinkers were patients who used to have an excessive alcohol consumption (i.e., exceeding the WHO recommendations) and had not consumed alcohol for more than one month. Never-drinkers were defined as having reported no alcohol consumption at any age.

Nicotine dependence in current and former smokers was assessed with the Fagerström score [6,21], a 6-item questionnaire with an overall score ranging from 0 to 8 (0 to 2 points: no dependence; 3 to 4 points: low dependence; 5 to 6 points: medium dependence; 7 to 8 points: high dependence). Following the guidelines for this questionnaire, patients with a Fagerström score > 2 were addressed to a tobacco specialist.

Alcohol abuse was assessed by the CAGE screening questionnaire [22], a 4-question clinical interview that has proved useful to diagnose excessive drinking and alcoholism. Patients with a positive answer to 2 or more of the 4 questions were considered as having a problematic drinking behavior and were addressed to an addictologist. Never-drinkers were not assessed.

Variation of nicotine dependence was defined as any variation in the Fagerström score. Variation of alcohol dependence was defined as any variation in the CAGE score.

Social deprivation was assessed using the EPICES score [23] (Evaluation de la Précarité des Inégalités de Santé dans les Centres d’Examens de Santé—Evaluation of Deprivation and Inequalities in Health Examination Centres). The EPICES score includes 11 binary items (yes/no) covering the dimensions of social and material deprivation shown to be strongly correlated with several health indicators [11]: marital status (one item), health insurance status (one item), economic status (three items), family support (three items), and leisure activity (three items). The score ranges from 0 (no deprivation) to 100 (maximal deprivation) with a threshold at 30.17 [11]. Patients with an EPICES score ≥ 30.17 were considered deprived and were addressed to the social worker.

All patients identified as alcohol- and/or nicotine-dependent (Fagerström score > 2 and/or ≥2 positive answers to the CAGE score) and/or deprived (EPICES score ≥ 30.17) at T0, T1, and/or T2 were referred to addiction care and/or social services according to their assessment. To facilitate patients’ adherence, the appointments were scheduled the same day as the medical appointments for cancer treatment.

Descriptive statistics were produced using Excel^®^ software (Microsoft corp, Redmond, WA, USA) and PRISM©. One-way ANOVA on ranks (Kruskal Wallis test) were used to evaluate continuous variables, and Fisher’s exact test was used to evaluate nominal variables. Predictive analysis was performed through univariate and multivariate logistic regression models using MEDCALC^®^ software. Independent variables with *p*-value < 0.20 were included in the multivariate model. A *p*-value < 0.05 was considered statistically significant.

## 3. Results

Overall, 329 patients were included in the study, and 182 patients alive 12 months after diagnosis (137 men and 45 women with a mean age of 64.3 (32–90) years old) were included in the present analysis. All eligible patients in this population had previously accepted to participate in the study.

Conversely, 147 patients presenting with HNSCC recurrence or who died before the 12-months follow-up were excluded from the present analysis.

Among the 182 patients included, HNSCC was located in the oral cavity for 63 patients (34.6%), in the larynx for 48 (26.4%), in the oropharynx for 43 (23.6%), and in the hypopharynx for 21 (11.5%). Seven patients had HNSCC of unknown primary (3.8%). The disease was stage I/II for 45 patients (25%) and stage III/IV for 137 patients (75%). Regarding patients’ family situations, 118 (64.8%) were living alone, 49 (26.9%) were marriedand 15 (8.3%) were living with a relative (i.e., family member or informal care giver). Regarding their activity, 108 (59%) were retired, 37 (20%) were employed, 19 (10%) were disabled, 16 (8.8%) were looking for a job, and 2 (1.1%) were under guardianship. The results at 6 months were excluded from the analysis due to too many missing data (21.4%).

### 3.1. At the Time of Diagnosis (T0)

#### 3.1.1. Comparative Analysis

Out of the 182 patients included in the present analysis, 92 (51%) patients were current smokers, 63 (35%) were former smokers, and 27 (15%) patients were never-smokers at diagnosis (Table 1). The mean number of pack-years for ever-smokers was 41 [±20.5], whereas the mean number of pack-years was 49 [±19.5] for current and 32 [±17.9] for former smokers. One current smoker refused to undergo the Fagerström test. The mean Fagerström score for current smokers was 4.09 [±2.67]. Among the 155 ever-smokers, 88 (57.1%) were retired, 33 (20.8%) were employed, 16 (10.4%) were disabled, 16 (10.4%) were unemployed, and 2 (1.2%) were under guardianship (Table 2). Among the 27 never-smokers, 20 were retired (74%), 4 were employed (14.8%), and 3 were disabled (11.1%) (Fisher, *p* = 0.3).

Regarding alcohol consumption, 78 (43%) patients were current drinkers, 38 (21%) were former drinkers, and 66 (36%) were never-drinkers at diagnosis (Table 3). Among the 116 ever-drinkers, 57 (49.1%) were retired, 28 (24.1%) were employed, 14 (12%) were disabled, 15 (12.9%) were looking for a job, and 2 (1.7%) were under guardianship (Table 4). Among the 66 never-drinkers, 51 (77.3%) were retired, 9 (12.2%) were employed, 5 (7.5%) were disabled, and 1 (1.5%) was unemployed (Chi2, *p* = 0.003).

Overall, the patients’ mean EPICES score was 29 [±20.9].

#### 3.1.2. Univariate Analysis

Current smokers were more frequently men (Fisher, *p* < 0.001), and were generally younger than former smokers and never-smokers (mean age = 60.9 [±9.4], 67.9 [±10.9], and 67.3 [±12.5] years for current, former, and never-smokers, respectively (*p* < 0.001)) (Table 1). Current smokers were also significantly more likely to be underweight (BMI < 18.5) than former or never-smokers (Fisher, *p* < 0.001) (Table 1). Current tobacco consumption was significantly associated with current alcohol consumption (Fisher, *p* < 0.001) (Table 1).

Likewise, current drinkers and former drinkers at baseline were more frequently men (Chi2, *p* < 0.001), and current drinkers (mean age = 61.8 [±9.41]) and former drinkers (mean age = 62.3 [±9.59]) were generally younger than never-drinkers (mean age = 68.3 [±12.3]) (Kruskall Wallis *p* < 0.001) (Table 3). Current drinkers tended to have a higher Fagerström score than former drinkers and never-drinkers (Kruskall Wallis, *p* < 0.001) (Table 3).

The mean EPICES score differed significantly by status: unemployed patients (mean EPICES score = 45.9 [±22.5]) and patients with an established guardianship (mean EPICES score = 54.1 [±18.8]) had a lower EPICES score than retired patients (mean EPICES score = 25 [±17.9]), employed patients (mean EPICES score = 31 [±23.2]), and disabled patients (mean EPICES score = 30.6 [±23.2]) (Kruskal-Wallis, *p* < 0.01) (Table 4).

The mean EPICES score varied significantly according to alcohol and tobacco consumption at diagnosis: current drinkers (mean EPICES score = 34.1 [±21.9]) and former drinkers (mean EPICES score = 32.7 [±21]) were more socially deprived than never-drinkers (mean EPICES score = 20.8 [±17.1]) (Kruskal-Wallis, *p* < 0.001); current smokers (mean EPICES score = 36.2 [±22.1]) were more socially deprived than former smokers (mean EPICES score = 23.6 [±18.7]) (Kruskal-Wallis, *p* < 0.001) (Table 4).

### 3.2. One Year after Diagnosis (T2)

#### 3.2.1. Comparative Analysis

One year after diagnosis, 38 (21%) patients were current smokers, 117 (64%) patients were former smokers, and 27 (15%) patients were never-smokers (Table 1). Among former smokers one year after diagnosis, 55 patients had quit smoking since diagnosis (i.e., 60% of current smokers at diagnosis). Among current smokers one year after diagnosis, 37 patients had been current smokers at baseline and one patient had been a former smoker at baseline and had started smoking again (Table 1).

The mean Fagerström score of persistent smokers decreased from 4.09 ± 2.67 at the time of diagnosis to 1.40 [±2.14] at t = 1 year. Among patients with nicotine dependence at baseline (Fagerström score > 2), the level of dependence decreased for 64 patients (41.5%), was stable for 6 patients (4%), and increased for 7 patients (4%) (Table 2).

Regarding alcohol consumption, 35 (19%) patients were current drinkers, 81 (44.5%) were former drinkers, and 66 (36%) patients were never-drinkers (Table 3). Among former drinkers at one year, 44 patients had stopped alcohol consumption since diagnosis (i.e., 56% of current drinkers at diagnosis). Among the 35 patients with alcohol consumption exceeding the WHO recommended threshold one year after diagnosis, 34 had been current drinkers at the time of diagnosis and one was a former alcoholic who relapsed.

Among ever-drinkers one year after diagnosis, 48 (41.7%) patients were less alcohol-dependent (CAGE score improvement), 8 (5.9%) patients had a stable alcohol dependence (stable CAGE score), and 6 (5.2%) patients had an increased alcohol dependence (worsened CAGE score) (Table 5).

Overall, the mean EPICES score one year after diagnosis was significantly improved 22.9 [±16.5] (Wilcoxon, *p* < 0.0001). Social deprivation decreased for 86 (47%) patients, was stable for 57 (32%) patients, and worsened for 38 (21%) patients (Table 4).

#### 3.2.2. Univariate Analysis

There was no statistically significant difference between patients who quit smoking at one year and those who continued regarding their age, their initial BMI, their sex, or the stage and tumor location.

Smoking cessation was significantly associated with alcohol cessation (Fisher, *p* < 0.001) (Table 1). Patients with decreased nicotine dependence (Fagerström score improvement) also experienced a decreased level of alcohol dependence (CAGE score improvement) (Fisher, *p* < 0.001) (Table 2). Patients with improved nicotine dependence were significantly younger (Kruskall-Wallis < 0.001) (Table 2). Retired and employed patients were also more likely to experience a decrease in their level of nicotine dependence than disabled and unemployed patients (Fisher, *p* < 0.02; Table 2).

Persistent drinkers and persistent smokers one year after diagnosis had a significantly higher EPICES score at the time of diagnosis (Kruskal-Wallis, *p* < 0.001) (Table 4).

Patients who continued to smoke one year after diagnosis were significantly more likely to continue to drink (*p* < 0.001) and had a significantly higher initial EPICES score (*p* < 0.001) (Table 1).

### 3.3. Predictive Analysis

Univariate logistic regression analysis with a significance level of 0.05 was performed in ever-smokers (Table 6) and found a statistically significant relationship between smoking cessation failure and the initial Fagerström score (OR = 1.632 [1.3852–1.9227]), alcohol cessation failure (OR = 5.4508 [2.3567–12.6069]) and social deprivation (EPICES score ≥ 30.17) (OR = 2.9514 [1.3865–6.2824]). This association remained in multivariate analysis, with a statistically significant relationship between tobacco cessation failure and the initial Fagerström score (OR = 1.6181 [1.3189–1.9851]) and alcohol cessation failure (OR = 5.3515 [1.7407–16.4528]).

Univariate logistic regression analysis with a significance level of 0.05 was performed in ever-drinkers and found a statistically significant relationship between alcohol cessation failure and the initial CAGE score (OR = 2.329 [1.6600–3.2677]) and smoking cessation failure (OR = 3.8235 [1.5942–1704]). This association remained in multivariate analysis, with a statistically significant relationship between alcohol cessation failure and the initial CAGE score (OR = 2.4935 [1.7067–3.6429]) and smoking cessation failure (OR = 8.1147 [2.3355–28.1949]).

## 4. Discussion

The present study prospectively assessed the systematic screening and orientation of current smokers and drinkers at HNSCC diagnosis and tobacco and alcohol cessation concomitantly. Patients’ orientation took place at the time of diagnosis, the timing of when clinical intervention is offered appearing to be a critical factor in smoking cessation efforts in Santi et al.’s study [13]. In our study, alcohol/tobacco consumption and patient social deprivation tended to improve one year after HNSCC diagnosis, when these frailties were screened for and assessed at the time of diagnosis. Predictive analysis underlined the relationship between alcohol and tobacco cessation failure, the dependence degree (measured with the Fagerström score and the CAGE score), and the EPICES score. These indicators appear to be good predictors of cessation failure and might help oncologists in selecting patients requiring a tailored intervention targeting addictions and/or social deprivation.

It is estimated that 14% to 59% of patients with an HNSCC will continue to smoke after their diagnosis [15], although 85.6% of them initially declare that they wish to quit [24]. Moreover, cigarette smoking is frequently associated with alcohol drinking [2,15]. However, tobacco and alcohol cessation programs are proposed to cancer patients by less than half of oncologists [25], caregivers focusing mainly on the pathology when treating cancer patients [16]. This underlines the necessity of a more patient-centered approach for patients enrolled in the cancer care pathway, including the management of addictions [16].

Although the heterogeneity of cessation assistance offered might play a role [14], the broad range of smoking (33–70%) [26] and alcohol cessation (13–24%) [12,27] rates reported among HNSCC patients suggests that smoking and drinking are affected by other factors than the cancer diagnosis itself in this population. Factors influencing the patterns of smoking include low income, poor housing, and unemployment [28]; nicotine exposure during childhood [29] and parental/peer example [30]; financial pressure and stress [31]; anxiety and depression [32]; efficient targeted marketing [33]; and outdoor working [34]. Likewise, vulnerability to alcohol abuse seems to be related to multiple genetic or environmental factors or their interaction [35]. Smoking and drinking behaviors are thus correlated to SES [2,36]. Patients in this study combining the three conditions of drinking, smoking, and social deprivation together seem to experience more difficulties in quitting drinking and smoking, pointing out the need for enhanced support for these patients. However, despite the fact that socioeconomic status is an independent risk factor for HNSCC [10] correlated with survival [5], the relationship between a patient’s social environment and smoking and alcohol consumption have not been adequately explored yet [11,26].

Apart from individual motivation, difficulties with quitting tobacco or alcohol increase with patients’ dependence. Alcohol and nicotine dependence in HNSCC patients is generally elevated, as in the present study [24]. Therefore, we used the Fagerström score and the CAGE score to assess each patient’s degree of dependence and to target assistance to patients exhibiting a high dependence score [37]. Vulnerability to addiction and high dependence scores might be explained by genetic, environmental, and psychological factors [38,39,40], as dependence reflects both a physiologic and a psychosocial craving [41]. As previously reported in the literature [12,15,18], tobacco consumption and alcohol drinking were frequently associated in our study, alcohol drinking being exceptionally isolated. Therefore, in the present study, both addictions were targeted simultaneously, although uncertainties remain in the literature regarding the optimal therapeutic sequence (concurrent, sequential, or not linked at all).

Moreover, we observed that concurrent alcohol and tobacco dependence evolved in a similar way. Current drinkers tend to have a higher Fagerström score than former drinkers and never-drinkers, suggesting addiction-prone personality traits in cases of concomitant alcohol and tobacco use. As previously reported by Kagishar et al. [26], patients who continued to smoke one year after diagnosis were also those who continue to drink, despite receiving addiction counselling. Addictions also seemed to worsen among the more precarious patients in the present study, social precarity appearing to be related to heavier addictions.

As a consequence, some authors state that people who both drink and smoke will benefit the most from effective interventions targeting both behaviors at the same time, because they have a multiplicative effect compared with targeting each behavior individually [12]. This point is emphasized in a systematic review evaluating two smoking interventions and one smoking + alcohol intervention in people with HNSCC. Out of the three randomized control trials analyzed, only the smoking + alcohol intervention reduced smoking prevalence [17]. In another study with a similar design to the present study, the tobacco withdrawal rate was lower (23.7%) compared to the present study, which could be related to the single management of tobacco withdrawal in this study without management of other addictions, nor of social deprivation [13].

In our study, addictions tended to improve at one year when taken care of from the diagnosis of HNSCC. However, it is unclear how much of the baseline or follow-up health behaviors and lifestyle changes in this study are unique to the HNSCC population studied. The risk of developing HNSCC rises with the number of cigarettes per day and the actual duration of tobacco consumption. It is also considered that the risk of HNSCC decreases proportionally to the length of time since cessation [42]. A retrospective study from 2021 evaluating a cohort of 117 current smokers with p16-negative HNSCC prospectively enrolled in a tobacco treatment program showed that, after adjustment for age, comorbidity, and site, abstinent stage I to II patients had a decreased risk of death compared to smoking stage I to II patients [43]. Therefore, the importance of smoking and drinking cessation should be stressed from the diagnosis of HNSCC to improve survival in this population. These observations emphasize the need for precision interventions targeting both behaviors when a patient is diagnosed with HNSCC.

The present study has several strengths, including its prospective design and the large number of HNSCC patients included. This is the first study to measure the evolution of dependence, its systematic management being offered in the period following treatment for HNSCC. The main caveats of this study consist of the absence of a control group, as it would have been unethical to diagnose addictions or social difficulties without taking care of them. Additionally, the one-year follow-up period didn’t enable us to evaluate the long-term evolution of addictions and social deprivation in that population. While all patients with social deprivation and/or smoking/alcohol dependence attended addiction care and/or social services, there was no assessment in the present study of attendance of follow-up visits. Moreover, the shortage of addiction care professionals constituted a barrier to systematically addressing all patients with alcohol consumption to the addiction care. We therefore prioritized patients categorized as heavy drinkers according to the WHO. The repetition of the same tests can also create a learning phenomenon which could represent another bias. Finally, the tests chosen are not necessarily the most accurate in identifying patients who need support for their abstinence—in particular the CAGE test may be insufficiently specific [44]—and extending addiction counselling to all patients regardless of their level of addiction could have further improved quitting outcomes.

## 5. Conclusions

To conclude, individualized standard care incorporating social support and management of cancer risk behaviors, in particular tobacco and alcohol consumption, should be systematically offered, especially to patients who are more vulnerable and might need extra support to quit smoking and drinking [45,46]. Cigarette smoking being often associated with alcohol consumption in the HNSCC population [2], treating comorbid smoking and drinking might increase smoking and alcohol cessation rate and may be more effective than treating these disorders separately [18]. Smoking and alcohol drinking have different patterns of associated variables in post-therapeutic HNSCC patients, which have to be taken into account for intervention design [36]. Precision interventions targeting both behaviors when a patient is diagnosed with HNSCC might improve survival in this population.

## Figures and Tables

**Table 1 cancers-15-01231-t001:** Univariate analysis according to tobacco consumption. (Former drinkers were patients who no longer drank but had been drinking before data cutoff).

At Time of Diagnosis (T0)							One Year after Diagnosis (T2)						
	Current smoker (*n* = 92)	Current non-smoker (*n* = 90)					Current smoker (*n* = 38)		Current non-smoker (*n* = 144)				
	Current smoker (*n* = 92)	Former smoker (*n* = 63)	Never-smoker (*n* =27)	*n*	*p*	Test	Persistent smoker (*n* = 37)	Smoking Relapse (*n* = 1)	Former smoker (*n* = 117)	Never-smoker (*n* = 27)	*n*	*p*	Test
Age (mean ± SD)	60.9 ± 9.5	67.9 ± 10.9	67.3 ± 12.5	182	<0.001	Kruskal-Wallis	60.4 ± 9.3	60	64.9 ± 10.8	67.3 ± 12.5	182	0.018	Kruskal-Wallis
**Sex**													
Male (*n* -%)	75 (54.7%)	54 (39.4%)	8 (5.8%)	137 (100%)	<0.001	Fisher	30 (21.9%)	1 (0.7%)	98 (71.5%)	8 (5.8%)	137 (100%)	<0.001	Fisher
Female (*n* -%)	17 (37.8%)	9 (20%)	19 (42.2%)	45 (100%)			7 (15.6%)	0	19 (42.2%)	19 (42.2%)	45 (100%)		
**Body Mass Index (BMI)**													
<18.5	17 (89.5%)	2 (10.5%)	0	19 (100%)	<0.001	Fisher							
18.5–24.9	49 (50%)	30 (30.6%)	19 (19.4%)	98 (100%)									
25–30	24 (54.7%)	25 (47%)	4 (7.5%)	53 (100%)									
>30	2 (16.6%)	6 (50%)	4 (33.3%)	12 (100%)									
**Addiction and Precarity**													
CAGE score (mean ± SD)	1.5 ± 1.4 1 missing value	0.7 ± 1.2	2 ± 1	115	0.008	Kruskal-Wallis	1.47 ± 1.63 (1 missing value)	0	1.13 ± 1.39	0	115	0.58	Kruskal-Wallis
EPICES score (mean ± SD)	36.2 ± 22.1 (2 missing values)	22.8 ± 17.8	18.9 ± 14.5	180 (2 missing values)	<0.001	Kruskal-Wallis	29.0 ± 22.1 (2 missing values)	49.1	20.7 ± 20.2	18.9 ± 14.5	180 (2 missing values)	0.002	Kruskal-Wallis
**Alcohol Consumption:**													
Current drinker (*n* -%)	55 (70.5%)	21 (26.9%)	2 (2.5%)	78 (100%)	<0.001	Fisher	18 (52.9%)	0	14 (41.2%)	2 (5.9%)	34 (100%)	<0.001	Fisher
Former drinker (*n* -%)	19 (50%)	18 (47.4%)	1 (2.6%)	38 (100%)			15 (18.5%)	1 (1.2%)	65 (80.2%)	1 (1.2%)	81 (100%)		
Never-drinker (*n* -%)	18 (27.3%)	24 (36.4%)	24 (36.4%)	66 (100%)			4 (6%)	0	38 (57.6%)	24 (36.4%)	66 (100%)		
Alcohol relapse (*n* -%)							1 (100%)	0	0	0	1 (100%)		

SD = standard deviation.

**Table 2 cancers-15-01231-t002:** Variation of the nicotine dependence according to the Fagerström score, monitored in 154/155 ever-smokers (willing to undergo the Fagerström test) at time of diagnosis and within one year from the diagnosis.

Variation of Nicotine Dependence from Baseline (T0) to One Year Post-Diagnosis (T2)								
			Never nicotine-dependent (*n* = 77)	Improved nicotine dependence (*n* = 64)	Stable nicotine dependence (*n* = 6)	Worsened nicotine dependence (*n* = 7)	*p*	Test
Age (mean ± SD)		154	67.2 ± 1.10	59.8 ± 8.94	64.2 ± 5.04	62.7 ± 12.7	<0.001	Kruskal-Wallis
**Sex**								
Male (*n* -%)		129	65 (84%)	55 (86%)	4 (67%)	5 (71%)	0.66	Fisher
Female (*n* -%)		25	12 (16%)	9 (14%)	2 (33%)	2 (29%)		
**According to EPICES Status at Time of Diagnosis**								
Mean EPICES SCORE (±SD)		152	22.6 ±16.2	39.3 ±23.7	39.0 ±19.8	36.0 ±20.1	0.02	Kruskal-Wallis
**According to Activity at Time of Diagnosis**								
Retired (*n* -%)		88 (57.1%)	53 (60.2%)	28 (31.8%)	4 (4.5%)	3 (3.5%)	0.02	Fisher
Employed		32 (20.8%)	14 (43.8%)	16 (50%)	1 (3.1%)	1 (3.1%)		
UnemployedE	Disabled (*n* -%)	16 (10.4%)	8 (50%)	7 (43.8%)	0	1 (6.2%)		
	Looking for a job (*n* -%)	16 (10.4%)	2 (12.5%)	12 (75%)	1 (6.25%)	1 (6.25%)		
	Guardianship (*n* -%)	2 (1.2%)	0	1 (50%)	0	1 (50%)		
**According to the Variation of Alcohol Dependence among Ever-Drinkers (N = 112)**								
Never alcohol-dependent (*n* -%)		50	30 (26%)	18 (16%)	2 (1.8%)	1 (0.9%)	<0.001	Fisher
Improved alcohol dependence (*n* -%)		48	14 (12.5%)	32 (28.4%)	0 (0%)	2 (1.8%)		
Stable alcohol dependence (*n* -%)		6	0	3 (2.7%)	2 (1.8%)	1 (0.9%)		
Worsened alcohol dependence (*n* -%)		8	2 (1.8%)	2 (1.8%)	1 (0.9%)	3 (2.7%)		

SD = standard deviation.

**Table 3 cancers-15-01231-t003:** Univariate analysis according to alcohol consumption.

		At Time of Diagnosis							One Year after Diagnosis			
		Current drinker (*n* = 78)	Current non-drinker (*n* = 104)				Current drinker (*n* = 35)		Current non-drinker (*n* = 147)			
	*n*	Current drinker (*n* = 78)	Former drinker (*n* = 38)	Never-drinker (*n* = 66)	*p*	Test	Persistent drinker (*n* = 34)	Alcohol relapse (*n* = 1)	Former drinker (*n* = 81)	Never-drinker (*n* = 66)	*p*	Test
Age (mean ± SD)	182	61.8 ± 9.4	62.3 ± 9.6	68.3 ± 12.3	<0.001	Kruskal-Wallis	60	62	61	69	<0.001	Kruskal-Wallis
**Sex**												
Male (*n* -%)	137	70 (38.5%)	30 (16.5%)	37 (20%)	<0.001	Fisher	32	0	68	37	<0.001	Fisher
Female (*n* -%)	45	8 (4.5%)	8 (4.5%)	29 (16%)			2	1	13	29		
**BMI**												
<18.5	98	42 (54%)	22 (58%)	34 (52%)	0.23	Fisher						
18.5–24.9	53	21 (27%)	9 (24%)	23 (35%)								
25–30	19	11 (14%)	6 (16%)	2 (3%)								
>30	12	4 (5.1%)	1 (2.6%)	7 (11%)								
**Addiction and Precarity**												
Fagerström score (mean ± SD)	154	3.24 ± 2.86	2.19 ± 3	1.14 ± 2.29	<0.001	Kruskal-Wallis	3.75 ± 3.1 (1 missing value)	0	2.58 ± 2.7	0.35 ± 1.1	<0.001	Kruskal-Wallis
EPICES score (mean ± SD)	180	34.1 ± 21.9 (1 missing value)	32.7.8 ± 21	20.8 ± 17.1 (1 missing value)	<0.001	Kruskal-Wallis	35.2 ± 22.1 (1 missing va1ue)	30.8	33.3 ± 21.6	15.9 ± 15.1 (1 missing value)	0.02	Kruskal-Wallis

SD = standard deviation.

**Table 4 cancers-15-01231-t004:** Univariate analysis according to social deprivation (EPICES score).

Years		EPICES Score (Mean ± SD)	*n*	*p*	Test
t = 0	Family situation				
	Living alone	23.6 ± 18.7 (2 missing values)	118 (64.8%)	<0.001	Kruskal-Wallis
	Married life/In a relationship	36.9 ± 20.9	49 (26.9%)		
	Living with a relative (i.e., family member or informal care giver)	45.3 ± 22.2	15 (8.3%)		
	Activity				
	Retired	25 ± 17.9 (1 missing value)	106 (58.2%)	<0.001	Kruskal-Wallis
	Employed	31 ± 23.2 (1 missing value)	39 (21.4%)		
	Disabled	30.6 ± 23.2	19 (10.4%)		
	Looking for a job	45.9 ± 22.5	16 (8.8%)		
	Guardianship	54.1 ± 18.8	2 (16.6%)		
	Body Mass Index (BMI)				
	<18.5	50.1 ± 23.3	19 (10.4%)	<0.001	Kruskal-Wallis
	18.5–24.9	28.3 ± 20.5	97 (53.3%) (2 missing value)		
	25–30	24.3 ± 16.9	53 (29.1%)		
	>30	21 ± 17.4	11 (6%)		
Years		EPICES score at time of diagnosis (mean SD)	*n*	*p*	Test
t = 1	Function of variation of the EPICES score				
	Improved EPICES score	39.8 ± 21.7	86 (47.2%)	<0.001	Kruskal-Wallis
	Stable EPICES score	16.2 ± 12.6	57 (31.3%)		
	Worsened EPICES score	23.8 ± 15.7	37 (20.3%)		
	Function of Tobacco status one year after diagnosis				
	Never-smoker	6.5 ± 11.6	27 (14.8%)	<0.001	Kruskal-Wallis
	Former smoker	12.42 ± 15.06	37 (20.3%)		
	Persistent smoker	45.6 ± 26.6	115 (63.2%) (2 missing values)		
	Smoking relapse	73.37	1 (0.5%)		
	Function of alcohol status one year after diagnosis				
	Never-drinker	6.1 ± 15.7	66 (36.3%) (1 missing value)	0.07	Kruskal-Wallis
	Former drinker	11.6 ± 13.6	81 (44.5%)		
	Persistent drinker	37.8 ± 28.8	33 (18.2%)		
	Drinking relapse	Missing data	1 missing value		

SD = standard deviation.

**Table 5 cancers-15-01231-t005:** Variation of alcohol dependence according to the CAGE score, assessed among 115/116 ever drinkers (willing to undergo the CAGE test) at the time of diagnosis and within one year from diagnosis.

	Variation of Alcohol Dependence								
Years			*n*	Never alcohol-dependent (*n* = 53)	Improved alcohol dependence (*n* = 49)	Stable alcohol dependence (*n* = 8)	Worsened alcohol dependence (*n* = 5)	*p*	Test
T = 1	Age		115	62.0 ± 9.6	61.0 ± 10.1	53.5 ± 11.7	62.0	0.29	Kruskal-Wallis
	EPICES score at diagnosis (mean ± SD)		114 (1 missing value)	28.8 ± 21.9	29.6 ± 18.3 (1 missing value)	35.8 ± 19.37	34.6 ± 35.9	0.43	Kruskal-Wallis
	Fagerström score (mean ± SD)		114 (1 missing value)	0.788 ± 1.66	0.617 ± 1.26	3.67 ± 2.25	3.50 ± 4.40 (1 missing value)	<0.001	Kruskal-Wallis
	**According to Activity at Time of Diagnosis**								
	Retired (*n* -%)		57 (49.1%)	27 (51%)	23 (47%)	3 (38%)	4 (80%)	NS 0.17	Fisher
	Employed (*n* -%)		28 (24.1%)	13 (25%)	12 (24%)	2 (25%)	1 (20%)		
	Unemployed	Disabled (*n* -%)	14 (12%)	7 (13%)	6 (12%)	1 (13%)	0		
		Looking for a job (*n* -%)	15 (12.9%)	6 (11%)	8 (16%)	1 (13%)	0		
		Guardianship (*n* -%)	2 (1.7%)	0	0	1 (13%)	0		

**Table 6 cancers-15-01231-t006:** Predictive analysis of determinants of persistent consumption in ever-smokers and ever-drinkers.

**Predictors of Smoking Persistence in Ever-Smokers**										
**Independent Variables**	**Univariate Analysis**					**Multivariate Analysis**				
	**Coefficient**	**Std.** **Error**	***p*-Value**	**OR**	**CI 95%**	**Coefficient**	**Std. Error**	***p*-Value**	**OR**	**CI 95%**
Activity	0.3186	0.1643	0.0524	1.3753	0.9967 to 1.897	0.07142	0.2758	0.7957	1.074	0.6255 to 1.8441
Sex	−0.1525	0.4878	0.7546	0.8586	0.3300 to 2.2337	−1.1672	0.6662	0.0798	0.3112	0.0843 to 1.1487
Age	−0.0427	0.01912	0.0256	0.9582	0.9229 to 0.9948	0.005972	0.03396	0.8604	1.006	0.9412 to 1.0752
Number of pack-years	0.02068	0.009016	0.0218	1.0209	1.0030 to 1.0391	0.007017	0.0148	0.6355	1.007	0.9782 to 1.0367
Alcohol cessation failure	1.6958	0.4278	0.0001	5.4508	2.3567 to 12.6069	1.6774	0.573	0.0034	5.3515	1.7407 to 16.4528
Fagerström score	0.4898	0.08366	<0.0001	1.632	1.3852 to 1.9227	0.4813	0.1043	<0.0001	1.6181	1.3189 to 1.9851
Evolution of EPICES score	0.2219	0.2413	0.3579	1.2484	0.7779 to 2.0035	−0.2246	0.3089	0.4671	0.7988	0.4360 to 1.4634
Initial EPICES score ≥ 30.17	1.0823	0.3854	0.005	2.9514	1.3865 to 6.2824	0.6755	0.502	0.1784	1.965	0.7346 to 5.2562
**Predictors of Drinking Persistence in Ever-Drinkers**										
**Independent variable**	**Univariate analysis**					**Multivariate analysis**				
	**Coefficient**	**Std. Error**	***p*-Value**	**OR**	**CI 95%**	**Coefficient**	**Std. Error**	***p*-Value**	**OR**	**CI 95%**
Activity	0.2523	0.1789	0.1585	1.2869	0.9063 to 1.8274	0.2785	0.3158	0.3777	1.3212	0.7115 to 2.4533
Sex	0.648	0.677	0.3384	1.9118	0.5072 to 7.2059	0.1547	0.8641	0.8579	1.1674	0.2146 to 6.3493
Age	−0.02737	0.02353	0.2446	0.973	0.9292 to 1.0189	0.03564	0.03948	0.3666	1.0363	0.9591 to 1.1196
Smoking cessation failure	1.3412	0.4463	0.0027	3.8235	1.5942 to 9.1704	2.0937	0.6354	0.001	8.1147	2.3355 to 28.1949
CAGE score	0.8455	0.1728	<0.0001	2.329	1.6600 to 3.2677	0.9137	0.1934	<0.0001	2.4935	1.7067 to 3.6429
Evolution of EPICES score	0.1241	0.2576	0.6301	1.1321	0.6833 to 1.8757	0.1628	0.3741	0.6634	1.1768	0.5653 to 2.4499
EPICES score ≥ 30.17	−0.1787	0.4306	0.6781	0.8364	0.3597 to 1.9449	0.675	0.675	0.0221	0.2134	0.0568 to 0.8013

Std. Error = standard error, OR = odds-ratio, CI = confidence interval.

## Data Availability

The data that support the findings of this study are available from the corresponding author, Sophie Deneuve, upon reasonable request.

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
