# Peer review of "Addictions, Social Deprivation and Cessation Failure in Head and Neck Squamous Cell Carcinoma Survivors"

_cancers, 2023, doi:10.3390/cancers15041231_

Round 1
Reviewer 1 Report
This study reports on longitudinal data exploring tobacco and alcohol use among head and neck cancer survivors in Lyon, France. This paper focuses on an important topic, and I appreciated the focus on both risk behaviors, given their synergist effect for cancer development and progression. However, the use of only univariate analyses and some confusion in the coding of variables included in the analysis reduced my enthusiasm. Specific comments follow.
Abstract – If possible, it would be more informative to show measures of association and not only p-values when describing associations.
Introduction – The introduction would be stronger if there was a longer presentation of previous work in this area. It is currently not clear what gap this study aims to fill. For example, paragraph 3 of the introduction talks about how fewer than half of oncologists are offering alcohol or tobacco treatment services, but this study does not address this aspect of cancer patient care. It seems to be assumed in the Methods that all patients in this sample were offered services. Is this known to be the case?
Methods -
How many patients who were approached declined to participate? This information is needed to be able to determine the generalizability of the patient sample.
Were the data collected by nurses at 6 and 12 months during in person visits or phone? What were the response rates for both follow-up periods?
Drinking categorization. It seems that drinkers who weren't heavy based on WHO guidelines are missing from this list of drinking categories. How was this group handled?
EPICES scores are described as, “Patients with an EPICES score ≤30.17, were considered deprived” but I think the authors mean that those with scores >= 30.17 were referred for follow-up care as increasing score represents worse deprivation.
The variable about change in nicotine dependence (Table 2) needs to be defined in the Methods.
Family status. There were no married patients with children? Why call out parents for one group but not others?
More details on the analyses were needed, what was compared? baseline to t1 and t2? characteristics across levels of behaviors? Also looks like nothing from 6 months is presented, why not?
The reliance on univariate statistics and multiple, unfocused comparisons makes it difficult to determine the key results from these analyses. It is unclear why only descriptive statistics are presented here when longitudinal data is available and the sample is large enough to build multivariable models. For example, the authors could have described the overall population at baseline and then explored predictors of t1 and t2 behaviors (i.e., change in tobacco use and alcohol behaviors) using multivariable analyses. Or, given that two time points are available, a mediation analysis looking at direct and indirect effects on outcomes could have been developed. At a minimum, the independent associations between demographic predictors and the outcomes of interest should be presented.
More information about what the referral to services are would be useful to put the results into context. Also, is there any information available on which patients used services that were offered to them?
Author Response
Reviewer 1 :
This study reports on longitudinal data exploring tobacco and alcohol use among head and neck cancer survivors in Lyon, France. This paper focuses on an important topic, and I appreciated the focus on both risk behaviors, given their synergist effect for cancer development and progression. However, the use of only univariate analyses and some confusion in the coding of variables included in the analysis reduced my enthusiasm. Specific comments follow.
Abstract – If possible, it would be more informative to show measures of association and not only p-values when describing associations.
Answer : Thank you for your comment. We used Fisher's exact, thus there is no measure of association to report. We now noted in the abstract that we used Fisher's exact test.For example, line 36/37 :
At the time of diagnosis, an active tobacco consumption was associated with alcohol drinking (Fisher’s exact, p<0.001).
Introduction – The introduction would be stronger if there was a longer presentation of previous work in this area. It is currently not clear what gap this study aims to fill. For example, paragraph 3 of the introduction talks about how fewer than half of oncologists are offering alcohol or tobacco treatment services, but this study does not address this aspect of cancer patient care. It seems to be assumed in the Methods that all patients in this sample were offered services. Is this known to be the case?
Answer : We thank you for these remarks. As suggested, a longer presentation of previous work in this area was added to the introduction.
Modifications are highlighted in yellow.
Methods: As stated line 135/136, all patients identified as precarious or dependent at T0, T1 and/or T2 were offered to be referred to alcohol or tobacco treatment services and/or social services.
We have now modified this sentence for further clarification (Line 135/137):
All patients identified as alcohol and/or nicotine dependent (Fagerström score > 2 and/or ≥ 2 positive answers to the CAGE score), and/or deprived (EPICES score ³ 30.17), at T0, T1 and/or T2, were referred to addiction care and/or social services according to their assessment. To facilitate patients' adherence, the appointments were scheduled the same day as the medical appointments for cancer treatment.
Methods -
How many patients who were approached declined to participate? This information is needed to be able to determine the generalizability of the patient sample.
Answer: All patients newly diagnosed with HNSCC, for whom surgical treatment was planned, accepted to participate in the study. No patients were lost to follow up, however, some patients refused to answer the tests, reported as missing variables in the tables (missing data).We chose to keep them in the analysis because most of the time, these patients refused a single test, and responded to others.
For further clarification, we now specified line 146/147:
Overall, three hundred and twenty-nine patients were included into the study, and 182 patients alive 12 months after diagnosis (137 men and 45 women with a mean age of 64.3 [32 - 90] years old), were included in the present analysis. All eligible patients in this population had previously accepted to participate in the study.
Were the data collected by nurses at 6 and 12 months during in person visits or phone? What were the response rates for both follow-up periods?
Answer : As specified lines 104-109 of the manuscript, the data were collected face to face by nurses at baseline and during follow-up consultations. We now specified that data were collected face to face.
Due to the high rate of missing data at 6 months (21.4%), these results were not included in the analysis. This low response rate might be due to the fact that patients are probably less prone to answer questions at the very end of their treatment due to fatigue. Most patients answered to questions at 1 year with very few missing data as shown in the tables.
Drinking categorization. It seems that drinkers who weren't heavy based on WHO guidelines are missing from this list of drinking categories. How was this group handled?
We thank the reviewer for this comment. Drinkers that weren’t heavy based on WHO guidelines were not addressed to addiction care. The high number of patients screened, and shortage of addiction care professionals did not allow to address all patients with alcohol consumption to the addiction care. To systematize management of patients included in the present study, we prioritized patients categorized as heavy drinkers according to WHO. We have added a sentence in the discussion (lines 817-819) to address this barrier:
Moreover, the shortage of addiction care professionals constituted a barrier to systematically address all patients with alcohol consumption to the addiction care. We therefore prioritized patients categorized as heavy drinkers according to WHO.
EPICES scores are described as, “Patients with an EPICES score ≤30.17, were considered deprived” but I think the authors mean that those with scores >= 30.17 were referred for follow-up care as increasing score represents worse deprivation.
Answer : Indeed, it was a typo, that was now corrected. Thank you for notifying it.
The variable about change in nicotine dependence (Table 2) needs to be defined in the Methods.
Answer : Nicotine dependence is assessed with the Fagerström score. Decreased nicotine dependence was defined as any decrease in the Fagerström score. This is now specified in the materiel and method section (line 125-126):
Variation of nicotine dependence was defined as any variation in the Fagerström score. Variation of alcohol dependence was defined as any variation in the CAGE score.
Family status. There were no married patients with children? Why call out parents for one group but not others?
Answer : We thank you for pointing out this inaccuracy. HNSCC patients are usually over 50 years old and are often isolated. Therefore, we wanted to evaluate support from relatives (i.e. family member or informal care giver) for these patients. It seems that we had difficulties to translate this idea correctly into english, because it did not seem clear to you.
We have modified the sentence lines 150/151:
Regarding patients’ family situation, 118 (64.8%) were living alone, 49 (26.9%) were living with a relative (i.e. family members or informal care giver), and 15 (8.3%) were married.
We also modified the table 4 accordingly.
More details on the analyses were needed, what was compared? baseline to t1 and t2? characteristics across levels of behaviors? Also looks like nothing from 6 months is presented, why not?
Answer: Alcohol /tobacco consumption and level of addiction, and social deprivation were assessed at baseline and T2. The 6 months results were not presented owing to too many missing data, patients being probably less prone to answer questions right after radiation therapy. We used the Fagerström score and the CAGE score to assess alcohol and tobacco addiction, and the EPICES score to evaluate social deprivation. The predictive analysis confirmed that the degree of alcohol/tobacco dependence assessed with CAGE/ Fagerström score was related to failure in alcohol/ tobacco cessation. These indicators could be used to offer the most dependent patients reinforced cessation support Besides, interestingly,
- A failure in alcohol cessation was linked with a failure in tobacco cessation (line 309), as well as a failure in tobacco cessation was linked with a failure in alcohol cessation in ever smokers
We added the results of the predictive analysis in the manuscript.
The reliance on univariate statistics and multiple, unfocused comparisons makes it difficult to determine the key results from these analyses. It is unclear why only descriptive statistics are presented here when longitudinal data is available and the sample is large enough to build multivariable models. For example, the authors could have described the overall population at baseline and then explored predictors of t1 and t2 behaviors (i.e., change in tobacco use and alcohol behaviors) using multivariable analyses. Or, given that two time points are available, a mediation analysis looking at direct and indirect effects on outcomes could have been developed. At a minimum, the independent associations between demographic predictors and the outcomes of interest should be presented.
Answer : Thank you for your comment and suggestion. Following your advice, we added the results of predictive analysis in the manuscript (line 300-309).
More information about what the referral to services are would be useful to put the results into context. Also, is there any information available on which patients used services that were offered to them?
Answer : All patients with an active tobacco and/or alcohol consumption were addressed to addiction care, and those with an EPICES score ³ 30.17 were addressed to a social worker. All patients went to their first appointment. This has now been specified line 142-145.
However, we did not collect data on patients’ follow-up with the addictologist and/or social worker. We added a sentence in the discussion pointing this as a limit of the study (line 817-819):
While all patients with social deprivation and or smoking/alcohol dependence attended addiction care and/or social services, there was no assessment in the present study of attendance of follow up visits.
The results of predictive analysis showed that the chosen indicators are good and might be used to select patients requiring an intervention. Moreover, the relationship between smoking and alcohol cessation failure is also underlined (line 309).
Reviewer 2 Report
It is with great pleasure I review the article, "Addictions, Social Deprivation and Cessation Failure in Head and Neck Squamous Cell Carcinoma Survivors" by Nokovitch et al., The authors should be commended for their work in this area and prospectively designing the study to determine the impact of smoking and alcohol cessation. In addition, the authors also evaluated the role of social factors in determining the survival of head and neck squamous cell carcinoma survivors. Apart from minor grammatical changes, I feel the article is well-written and the study was well-designed. No other comments from my end.
Author Response
Reviewer 2 :
It is with great pleasure I review the article, "Addictions, Social Deprivation and Cessation Failure in Head and Neck Squamous Cell Carcinoma Survivors" by Nokovitch et al., The authors should be commended for their work in this area and prospectively designing the study to determine the impact of smoking and alcohol cessation. In addition, the authors also evaluated the role of social factors in determining the survival of head and neck squamous cell carcinoma survivors. Apart from minor grammatical changes, I feel the article is well-written and the study was well-designed. No other comments from my end.
We thank the reviewer for his interest for our work.
Reviewer 3 Report
The article “Addictions, Social Deprivation and Cessation Failure in Head and Neck Squamous Cell Carcinoma Survivors” is very interesting and I have some comments to attempt to improve the article.
Materials and methods:
- You should include a bibliographical reference that provides the definition of tobacco as you did with alcohol (WHO, [14])
Limitations:
- You should point out as a limitation of the investigation the absence of a multivariate study that would determine the strength of association of the variables analyzed.
- Table 1. Reflect if ± is standard deviation. Add the percentage to the last row of Bodi mass index (n=2). Addiction and “precarity”: 49.1, does it have ±?
- Table 2: Worsened alcohol dependence (7?)
- Minor error. Table 1: never drinker (36.4%); Table 2, sex female (2, 33%)
Comment: Do you prefer to use “precarity” instead of precarious or precariousness?
Thank you
Author Response
Reviewer 3 :
The article “Addictions, Social Deprivation and Cessation Failure in Head and Neck Squamous Cell Carcinoma Survivors” is very interesting and I have some comments to attempt to improve the article.
Materials and methods:
- You should include a bibliographical reference that provides the definition of tobacco as you did with alcohol (WHO, [14])
Answer: Thank you for pointing out this imprecision. A reference was added.
Limitations:
- You should point out as a limitation of the investigation the absence of a multivariate study that would determine the strength of association of the variables analyzed.
Answer: Thank you for pointing this out. We added the results of two predictive analysis using persistance of tobacco consumption and persistence of alcohol consumption as dependant variables.
- Table 1. Reflect if ± is standard deviation. Add the percentage to the last row of Bodi mass index (n=2). Addiction and “precarity”: 49.1, does it have ±?
- Table 2: Worsened alcohol dependence (7?)
- Minor error. Table 1: never drinker (36.4%); Table 2, sex female (2, 33%)
Answer: Thank you for rising these issues. There was a layout problem making results less clear.This has been modified in the revised manuscript.
Comment: Do you prefer to use “precarity” instead of precarious or precariousness?
Thanks for this suggestion. We double checked, both terms are used interchangeably (400 occurrences in title and abstract in pubmed for “precarity” and 503 for “precariousness”).